# Nuciferine Inhibits Oral Squamous Cell Carcinoma Partially through Suppressing the STAT3 Signaling Pathway

**DOI:** 10.3390/ijms241914532

**Published:** 2023-09-26

**Authors:** Ji-Rong Xie, Xiao-Jie Chen, Gang Zhou

**Affiliations:** 1State Key Laboratory of Oral & Maxillofacial Reconstruction and Regeneration, Key Laboratory of Oral Biomedicine Ministry of Education, Hubei Key Laboratory of Stomatology, School & Hospital of Stomatology, Wuhan University, Wuhan 430079, China; xiejirong@whu.edu.cn (J.-R.X.); chenxiaojie-whu@whu.edu.cn (X.-J.C.); 2Department of Oral Medicine, School & Hospital of Stomatology, Wuhan University, Wuhan 430079, China

**Keywords:** cell-derived xenograft, nuciferine, oral squamous cell carcinoma, signal transducer and activator of transcription 3, tumor growth

## Abstract

Oral squamous cell carcinoma (OSCC) poses a significant obstacle to the worldwide healthcare system. Discovering efficient and non-toxic medications is crucial for managing OSCC. Nuciferine, an alkaloid with an aromatic ring, is present in the leaves of *Nelumbo nucifera*. It has been proven to play a role in multiple biological processes, including the inhibition of inflammation, regulation of the immune system, formation of osteoclasts, and suppression of tumors. Despite the demonstrated inhibitory effects of nuciferine on different types of cancer, there is still a need for further investigation into the therapeutic effects and potential mechanisms of nuciferine in OSCC. Through a series of in vitro experiments, it was confirmed that nuciferine hindered the growth, movement, and infiltration, while enhancing the programmed cell death of OSCC cells. Furthermore, the administration of nuciferine significantly suppressed the signal transducer and activator of transcription 3 (STAT3) signaling pathway in comparison to other signaling pathways. Moreover, the activation of the STAT3 signaling pathway by colivelin resulted in the reversal of nuciferine-suppressed OSCC behaviors. In vivo, we also showed the anti-OSCC impact of nuciferine using the cell-based xenograft (CDX) model in nude mice. Nonetheless, colivelin diminished the tumor-inhibiting impact of nuciferine, suggesting that nuciferine might partially impede the advancement of OSCC by suppressing the STAT3 signaling pathway. Overall, this research could offer a fresh alternative for the pharmaceutical management of OSCC.

## 1. Introduction

Oral squamous cell carcinoma (OSCC) is the most common histological type of head and neck cancer, accounting for 90% of all cases [1,2,3]. During the past 10 years, despite the extensive use of surgery, chemotherapy, radiation, and immunotherapy, the 5-year survival rate of OSCC patients has remained low [2,4]. Chemotherapy has a crucial role in OSCC treatment among the aforementioned traditional methods. Cisplatin, 5-FU, and paclitaxel are currently employed as the primary drugs for treating OSCC in the initial stage [5,6]. Despite the potential of these agents to decrease tumor size and enhance survival rates, the majority of OSCC patients eventually develop resistance to the medications through various unknown mechanisms, leading to relapse and unfavorable outcomes [7]. However, these medications frequently come with various adverse reactions, and the necessary therapeutic dosages are typically excessively harmful for patients to endure [7,8,9]. Therefore, it is imperative to search for efficient and nontoxic natural medications for the management of OSCC.

Nuciferine, an alkaloid containing an aromatic ring, is present in the leaves of *Nelumbo nucifera* [10,11]. For approximately a millennium in China, the leaves of the lotus plant (*N. nucifera*) have been recognized for their ability to combat obesity [12]. According to recent research, nuciferine has been found to exhibit a diverse array of additional effects. For instance, nuciferine has the ability to impact different cellular processes like cell growth, specialization, movement, self-degradation, and programmed cell death [12,13]. In the meantime, it participates in various biological processes, such as the creation of osteoclasts, the suppression of inflammation, the regulation of the immune system, and the prevention of tumor growth. Additionally, it is associated with several conditions, such as obesity, acute lung and kidney damage, and cancer [12,14,15,16,17,18]. In particular, nuciferine exhibited remarkable inhibitory effects on several types of cancer, including breast cancer, skin cutaneous melanoma, and lung cancer [11,19,20,21]. Furthermore, the non-synthetic compounds and bioactive substances derived from flora (like nuciferine) exhibit lower levels of toxicity, leading to a growing interest in their potential for cancer therapy [22,23]. However, despite the demonstrated inhibitory effects of nuciferine on different types of cancer, the therapeutic possibilities and the mechanisms by which nuciferine acts in OSCC remain unclear.

The objective of the research was to determine the influence of nuciferine on the behaviors of OSCC cells and the growth of tumors, along with investigating the underlying mechanisms associated with it. The study initially focused on examining the impact of nuciferine on the viability, inhibition rate, and proliferation of OSCC cells. Subsequently, the influence of nuciferine on the behavior of OSCC cells was individually evaluated. Moreover, the cell-derived xenograft (CDX) model provided evidence of nuciferine’s anti-tumor impact on the growth of OSCC. Finally, exploration of the signal transducer and activator of transcription 3 (STAT3) signaling pathway was undertaken. In summary, our research validated that nuciferine plays a repressive role in the functions and growth of OSCC cells, partly by inhibiting the STAT3 signaling pathway. This discovery offers a fresh therapeutic possibility for OSCC.

## 2. Results

### 2.1. Nuciferine Inhibited the Proliferation and Promoted the Apoptosis of OSCC Cells

In order to evaluate the cytotoxicity of nuciferine, a CCK-8 assay was carried out. The growth-inhibiting effects and cytotoxicity were detected in SCC25 and CAL27 cells treated with nuciferine (Figure 1A). The lowest concentration of NF inhibiting the proliferation of OSCC cells was 80 μM. Interestingly, nuciferine showed no viability-inhibiting effect on the normal cell HUVEC (Appendix A). Further dose-dependent viability tests were conducted with different nuciferine concentrations (ranging from 0 to 120 μM) over the duration of 24, 48, 72, and 96 h. According to the findings, nuciferine could suppress the proliferation of OSCC cells in a dose-dependent manner (Figure 1B and Appendix A). Moreover, after nuciferine treatment, a substantial reduction in cell proliferation was measured by colony formation (Figure 1C,D) and EdU assays (Figure 1E,F).

Apoptosis was triggered in OSCC cells as a result of being exposed to nuciferine for 48 h. Western blot analysis showed that the expression of pro-apoptosis markers Bax and Cleaved-caspase 3 was increased, while the expression of anti-apoptosis marker Bcl-2 was decreased in OSCC cells treated with nuciferine, implying that apoptosis induced by nuciferine was driven by the intrinsic apoptosis pathway (Figure 1G). Furthermore, flow cytometry results showed that the percentage of apoptotic cells in SCC25 and CAL27 cells increased significantly after being treated with nuciferine (Figure 1H,I).

### 2.2. Nuciferine Inhibited the Migration and Invasion of OSCC Cells

Wound-healing experiments were utilized to determine whether nuciferine inhibited the motility of OSCC cells. A low concentration of nuciferine (30 μM) was employed to study its effect on cell motility. Compared to the nuciferine-free control cells, nuciferine slowed the motility of OSCC cells at a low concentration that did not affect the cell proliferation, which were able to migrate and heal the scratch (Figure 2A,B). Similarly, migration of OSCC cells treated with 30 μM nuciferine was clearly reduced in transwell experiments (Figure 2C,D).

Transwell assays were used to assess the inhibitory effect of nuciferine on the invasion of OSCC cells. The results showed that the invasive ability of OSCC cells in the nuciferine treatment group was weakened compared to the control group (Figure 2E,F). Immunofluorescence results showed that the expression of E-cadherin increased while the expression of Vimentin decreased in OSCC cells after being treated with nuciferine (Figure 2G,H).

### 2.3. STAT3 Signaling Pathway Was Suppressed in Nuciferine-Treated OSCC Cells

The p65 and STAT3 signaling pathways were taken into consideration to explore the underlying signaling that mediated nuciferine-suppressed OSCC cell functions. By Western blot, the p65 and STAT3 signaling pathways were both found to be inhibited in OSCC cells after nuciferine treatment, in which, the suppression of the STAT3 signaling pathway was more obvious (Figure 3A). Furthermore, immunofluorescence staining results confirmed the STAT3 signaling-suppressive effect of nuciferine (Figure 3B).

### 2.4. Nuciferine Inhibited the Proliferation and Promoted the Apoptosis of OSCC Cells via Suppressing the STAT3 Signaling Pathway

Then, the involvement of the STAT3 signaling pathway was investigated. Colivelin was applied to activate the STAT3 signaling pathway suppressed by nuciferine. The EdU assay (Figure 4A,B) and the CCK-8 test (Figure 4C) both showed that nuciferine-inhibited proliferation of OSCC cells was reversed by colivelin. For apoptosis, flow cytometry results showed that the percentage of apoptotic cells in nuciferine-treated OSCC cells decreased when the STAT3 signaling pathway was activated (Figure 4D,E). Further, nuciferine treatment-increased expression of Bax and Cleaved-caspase 3 was downregulated, while nuciferine treatment-decreased expression of Bcl-2 was upregulated after colivelin treatment, indicating nuciferine-promoted OSCC cell apoptosis was impaired (Figure 4F).

### 2.5. Nuciferine Inhibited the Migration and Invasion of OSCC Cells via Suppressing the STAT3 Signaling Pathway

Next, we wondered whether the STAT3 signaling pathway also mediated nuciferine-inhibited OSCC cell migration and invasion. By wound healing assay, it was demonstrated that the nuciferine-slowed migration rate was accelerated after activating the STAT3 signaling pathway (Figure 5A,B). By a transwell assay, the invaded cell number of nuciferine-treated OSCC cells was increased by colivelin (Figure 5C,D). Moreover, increased expression of E-cadherin and decreased expression of Vimentin in nuciferine-treated OSCC cells were reversed after colivelin stimulation (Figure 5E,F).

### 2.6. Nuciferine Inhibited OSCC Growth via Suppressing the STAT3 Signaling Pathway

Finally, the effect of nuciferine on OSCC growth in vivo, and the involvement of the STAT3 signaling pathway were investigated by the CDX model in nude mice. The experimental procedure was shown in Figure 6A. After 2 weeks of drug injection, the mice were sacrificed. The results demonstrated that nuciferine could significantly suppress the growth of OSCC in vivo, with a lower tumor volume measured orthotopically (Figure 6B,D). Moreover, isolated tumors in the nuciferine group showed a lower tumor weight compared with the control group (Figure 6C,E). The results of the dynamic recording of tumor volume were consistent with the above findings (Figure 6F,G). However, when colivelin was co-injected intraperitoneally, nuciferine-suppressed OSCC growth was obstructed with increased tumor volume and weight during the process (Figure 6B–G). These results imply that nuciferine could inhibit OSCC growth partially via suppressing the STAT3 signaling pathway in vivo.

## 3. Discussion

Originally demonstrated to have a weight loss effect in traditional Chinese medicine, nuciferine is a significant alkaloid found in the leaves of *Nelumbo nucifera* [12]. Besides this, many studies showed that nuciferine exhibited suppressive effects on different kinds of cancer [11,21]. Hence, to ascertain the role of nuciferine in inhibiting OSCC, numerous experiments were performed to examine how nuciferine affects the viability, growth, programmed cell death, movement, and infiltration of SCC25 and CAL27 cells; the diagram below demonstrates the concept (Figure 7). According to the findings, nuciferine demonstrated the ability to suppress the malignant characteristics of OSCC cells. In the CDX model, the effectiveness of nuciferine in inhibiting OSCC growth was verified. Mechanismly, the suppressed STAT3 signaling pathway mediated nuciferine-inhibited OSCC cell functions and tumor growth. In summary, our research revealed that nuciferine has the ability to inhibit the growth of OSCC by impeding the harmful actions of OSCC cells, partly by suppressing the STAT3 signaling pathway.

Furthermore, we investigated the potential signaling pathways implicated in OSCC cells treated with nuciferine. Among these, the p65 and STAT3 pathways were confirmed to be compromised. Prior research has revealed that nuciferine has the ability to activate diverse signaling pathways in distinct cells, including AMPK, MAPK, PI3K/AKT, PPAR, NLRP3, and p65 signaling pathways [14,18,19,21,24]. Xu et al. studied the p65 signaling pathway. In a study, it was found that nuciferine exhibited inhibitory properties on the p65 signaling pathway in melanoma cells by specifically targeting Toll-like receptor 4 (TLR4). Consequently, this led to the suppression of melanoma cell growth and a reduction in tumor size [19]. Similarly, Li and colleagues verified that nuciferine has a significant effect on reducing the levels of Tripartite motif-containing 44 (TRIM44), leading to the inhibition of cell growth in laryngeal squamous cell carcinoma by suppressing the expression of phosphorylated p65 [25]. For the STAT3 signaling pathway, little work has focused on its involvement in nuciferine-suppressed tumor progression.

Conversely, multiple studies have indicated that the OSCC cells exhibited notable elevation in the phosphorylation of p65 and STAT3 [26,27,28,29]. The anti-cancer effect was observed when the p65 and STAT3 signaling pathways were inhibited, leading to the suppression of malignant behaviors in OSCC cells [30,31]. For instance, the study conducted by Lee and colleagues reported that the compound cudraxanthone H, which is derived from *Cudrania tricuspidata*, exhibited inhibitory effects on cell proliferation and induced apoptosis in OSCC cells by suppressing the p65 signaling pathway [32]. Sun et al. stated that the inhibition of Oct4 resulted in a reduction in inflammation and lowered levels of phosphorylation in p65 and STAT3. As a result, cell migration and invasion facilitated by lipopolysaccharide (LPS) and tumor necrosis factor-α (TNF-α) [33], were impaired. Based on previous research, our study revealed that nuciferine plays a part in repressing the p65 and STAT3 signaling pathways in OSCC cells, potentially accounting for the suppression of the malignant actions of OSCC cells.

Because the STAT3 signaling pathway was the most prominently inhibited pathway, our attention was ultimately directed towards investigating the role of the STAT3 signaling pathway in the inhibition of OSCC progression by nuciferine. Through a sequence of functional experiments conducted in vitro and utilizing an in vivo CDX model in nude mice, we discovered that the progression and growth of OSCC were hindered by nuciferine. However, this hindrance was exacerbated when the STAT3 signaling pathway was activated through the administration of a colivelin agonist. The findings suggested that nuciferine may exert its anti-OSCC effects by inhibiting the tumor-promoting STAT3 signaling pathway to some extent.

Further, in treating colon cancer and neuroblastoma, the anti-tumor working concentration of nuciferine reached 200 ug/mL in vitro, while the concentration used in animal experiments was 9.5 mg/kg [34]. In our study, the drug concentration for OSCC treatment was 80 μM (23 ug/mL) in vitro and 20 mg/kg in vivo. In terms of the difference in nuciferine concentration used in vitro and in vivo, we think the main reason is that the experimental environment in vitro is relatively single, while the in vivo environment is complex. Nuciferine may inhibit tumor growth by affecting the immune cells and stroma in the tumor microenvironment in vivo, leading to an obvious anti-tumor effect even with a low concentration.

Collectively, our research confirmed that nuciferine has the ability to hinder the growth, movement, and invasion of OSCC cells, while also enhancing their programmed cell death. Additionally, it restrains the advancement and development of OSCC in mice lacking a functional immune system. Furthermore, the inhibitory effects of nuciferine on malignant behaviors and tumor growth in OSCC were shown to be attributed to the suppression of the STAT3 signaling pathway. The aforementioned discoveries could potentially offer a fresh alternative for the pharmaceutical management of OSCC.

## 4. Materials and Methods

### 4.1. Reagents Antibodies and Cell Culture

For this research, the human tongue squamous cell carcinoma cell lines SCC25 and CAL27 were initially purchased from the American Type Culture Collection (Manassas, VA, USA). SCC25 cells were grown in DMEM/F-12 (Gibco, Grand Island, NY, USA), a mixture of Dulbecco’s Modified Eagle’s Medium and Nutrient Mixture F-12, with the addition of 10% FBS (fetal bovine serum) and 400 ng/mL hydrocortisone (Aladdin, Shanghai, China). The CAL27 cells were grown in DMEM (Gibco) supplemented with 10% FBS. HUVEC cells were maintained in 10% FBS DMEM. The cell lines were kept in a typical humidified incubator at 37 °C, with a 5% CO_2_ atmosphere. Plant Chem Medicine (Shanghai, China) provided nuciferine (PCM5266S). Dimethyl sulfoxide (DMSO) was used to dissolve a 10 mM stock solution of nuciferine. Colivelin was purchased from LEONBIO (Nanjing, China). The St John’s Laboratory was where the p-STAT3 antibody was acquired. Abcam was the source of the purchased Vimentin antibody. Abmart supplied antibodies for p-p65, Bax, and Bcl-2. Cell Signaling Technology was the source of the purchased antibodies for E-cadherin, Caspase 3, total-STAT3, and total-p65. Proteintech provided antibodies for β-actin, as well as horseradish peroxidase (HRP)-conjugated secondary antibodies, either goat anti-rabbit or goat anti-mouse.

### 4.2. Cell Counting Kit-8 (CCK-8) Assay

The cytotoxicity of nuciferine was examined using the CCK-8 assay. 4000 cells were cultured in each well of a 96-well plate and incubated in a humidified chamber at 37 °C for 24 h. The viability of the cells was assessed after stimulation with nuciferine (100 μM) for 24 h, followed by exposure to different concentrations of nuciferine (0, 40, 80, 120 μM) for various durations (0, 24, 48, 72, 96 h). After 24 h, the medium containing nuciferine was replaced with the complete medium and refreshed every 24 h. Additionally, colivelin (10 μM) was administered to examine the impact of the STAT3 signaling pathway on the inhibition of OSCC cell proliferation by nuciferine using CCK-8 (1, 3, 5 days). The CCK-8 assay was performed following the guidelines provided by the manufacturer (Dojindo, Tabaru, Japan), and the determination was based on the optical density (OD) reading at 450 nm.

### 4.3. Plate Colony-Forming Assay

To summarize, 500 OSCC cells were seeded in 6-well dishes for 1 day. The OSCC cells were exposed to nuciferine (80 μM) for an additional 24 h, after which the culture medium was replaced with either 10% FBS DMEM or DMEM/F-12. Following a week of cultivation, the cells were immobilized using 4% paraformaldehyde and subjected to staining with 0.5% crystal violet (Servicebio, Wuhan, China). Pictures were captured, and the quantity of cellular colonies was tallied.

### 4.4. Ethynyl-2-Deoxyuridine (EdU) Assay

The experiment to measure cell proliferation using EdU was conducted using the BeyoClickTM EdU Cell Proliferation Kit (Alexa Fluor 594; Beyotime, Nanjing, China). In short, OSCC cells were cultured in 48-well dishes with a cell density of 5 × 10^4^ cells per well. On the following day, a solution containing 30 μM of nuciferine was introduced and incubated for an additional 24 h. Additionally, an EdU assay (24 h) was conducted using colivelin (10 μM) to investigate the impact of the STAT3 signaling pathway on the inhibition of OSCC cell proliferation by nuciferine. After being double-stained with EdU and DAPI solutions following the manufacturer’s instructions, the EdU-positive cells were analyzed using the fluorescence microscope and quantified using ImageJ 1.8.0.

### 4.5. Flow Cytometry

For adhesion, OSCC cells were seeded overnight in 6-well plates at a density of 2.5 × 10^5^ cells per well. Following a 2-day period of nuciferine treatment (80 μM), OSCC cells underwent staining using the Annexin V-APC/PI Apoptosis Detection kit (Elabscience, Wuhan, China) as per the instructions provided by the manufacturer. To investigate the role of the STAT3 signaling pathway in promoting apoptosis of OSCC cells by nuciferine, colivelin at a concentration of 10 μM was administered for 48 h, and Annexin V-FITC/PI staining was used. Apoptotic cells were identified using flow cytometry.

### 4.6. Wound-Healing Assay

The scratches were created by using a 200 μL pipette tip on a monolayer of OSCC cells, followed by exposure to nuciferine (30 μM) in the presence of 1% FBS. To assess the migration of cells in wound closure, photographs were captured using a microscope at 0, 24 (SCC25), or 36 h (CAL27). To investigate the role of the STAT3 signaling pathway in the inhibition of OSCC cell migration by nuciferine, we also treated the cells with colivelin (10 μM) and performed a wound-healing assay at 0 and 24 h. ImageJ was utilized to calculate the distance of migration.

### 4.7. Transwell Assay

To evaluate the migratory and invasive potential of OSCC cells, a 24-well transwell system from Corning with an 8-μm pore size was employed. For the subsequent experiments, OSCC cells were incubated with nuciferine (30 μM) in 6-well plates for a duration of 24 h. In the transwell assay, the lower chamber was supplemented with medium containing 10% FBS, while the upper chamber was seeded with 1 × 10^5^ pre-treated cells in a medium also containing 10% FBS. To conduct the invasion test, the upper transwell chambers were coated with 100 μL of Matrigel (ABW; 082704, Shanghai, China) to establish a seamless membrane. In the upper chamber, the serum-free medium was inoculated with 5 × 10^5^ cells, while the lower chamber was supplemented with a medium containing 20% FBS. To investigate the role of the STAT3 signaling pathway in the inhibition of OSCC cell invasion by nuciferine, colivelin at a concentration of 10 μM was also utilized. After a 24 h incubation period, the upper chambers, housing cells that had migrated and invaded, were treated with 0.5% crystal violet stain. To eliminate the remaining cells on the upper surface of the membrane, a cotton swab was employed. Random photographs were taken under a microscope and quantified by ImageJ 1.8.0.

### 4.8. Western Blot

Proteins were obtained from OSCC cells by utilizing M-PER mammalian protein extraction reagent (Thermo Scientific, Waltham, MA, USA), and protein quantification was performed using a BCA kit (Beyotime, Nanjing, China). Afterward, 30 micrograms of protein from each group underwent electrophoresis using a 12% SDS-PAGE gel and were then transferred onto polyvinylidene difluoride (PVDF) membranes (Millipore, Billerica, MA, USA) using the wet transfer technique. After being blocked with 5% skimmed milk for 1 h, the membranes were incubated with primary antibodies overnight at 4 °C. After incubating with secondary antibodies at room temperature the following day, the membranes were observed using the Odyssey LI-COR scanner after being treated with the SuperSignal™ West Femto Reagent (Thermo Scientific, Waltham, MA, USA).

### 4.9. Immunofluorescence

For adhesion, 2.5 × 10^4^ cells were seeded in 48-well plates. The detection of STAT3 signaling change was observed after 48 h of treatment with nuciferine at a concentration of 80 μM. To investigate the role of the STAT3 signaling pathway in the inhibition of OSCC cell invasion by nuciferine, colivelin (10 μM) was used to stimulate OSCC cells pretreated with nuciferine for 24 h. The cells were then fixed with 4% paraformaldehyde for 10 min and permeabilized with 0.2% Triton X-100 (Beyotime, Nanjing, China) for 5 min. Following overnight incubation with primary antibodies targeting STAT3, Vimentin, and E-cadherin, the cells were subsequently exposed to fluorescent secondary antibodies for 1 h. Finally, DAPI (Beyotime, Nanjing, China) was applied for a duration of 5 min. A microscope was used to observe and capture photographs of cells.

### 4.10. Cell-Derived Xenograft (CDX) Tumor Model

In order to examine the impact of nuciferine on OSCC in living organisms and explore the connection between nuciferine-suppressed OSCC growth and the STAT3 signaling pathway, we created xenograft tumors in female BALB/c nude mice, aged 5 weeks, which were obtained from the Model Animal Research Center of Wuhan University. A total of six nude mice per group were subcutaneously implanted with 5 million CAL27 cells. Following a period of 10 days, nuciferine (20 mg/kg; dissolved in a solution containing 2% DMSO, 40% PEG400, 5% Tween-80, and 53% saline) and colivelin (1 mg/kg; dissolved in saline) were administered through intraperitoneal injections every 2 days. Following a 14-day period of drug administration, mice were euthanized using an excessive amount of anesthesia and subsequently captured in photographs. Using a Vernier caliper, the tumors were measured for their length (L) and width (W), and the tumor volume (V) was determined using the formula V = L × W^2^ × 0.5. The tumors were captured in pictures and measured in weight, and subsequently preserved in 4% paraformaldehyde. The Animal Protection and Use Committee of Wuhan University (WP20230091) granted approval for all experimental procedures.

### 4.11. Statistical Analysis

The data was analyzed using GraphPad Prism 8. The statistical analyses yielded a summary of the mean ± standard deviation (SD) from at least three independent experiments. The normal distribution was determined using the Shapiro-Wilk test. To assess the disparities, the *t*-test for Student’s, along with the one-way ANOVA incorporating Bonferroni correction, was utilized. The meaning of significance was determined as * *p* < 0.05, ** *p* < 0.01, *** *p* < 0.001 and **** *p* < 0.0001.

## Figures and Tables

**Figure 1 ijms-24-14532-f001:**
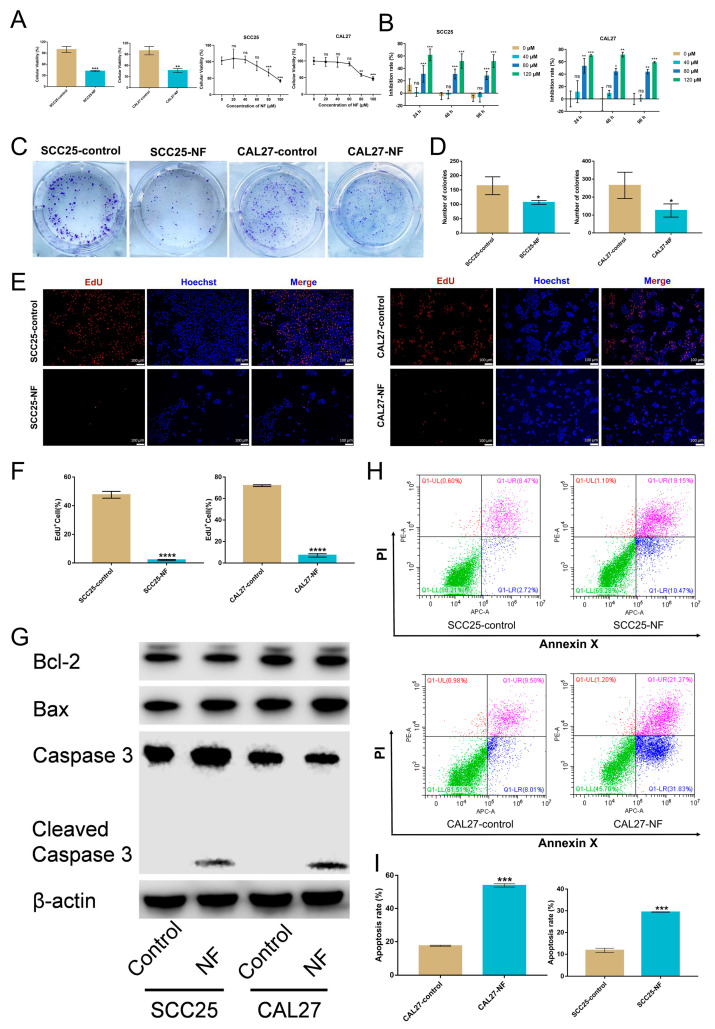
Nuciferine (NF) inhibited the proliferation and promoted the apoptosis of OSCC cells. (**A**) The growth-inhibiting effects of NF (100 μM, 24 h) and the cytotoxicity test of NF (0~100 μM, 24 h) on OSCC cells were detected by CCK–8. (**B**) The viability of SCC25 and CAL27 cells was inhibited by NF (0~120 μM, 0~96 h) in a dose-dependent manner, as verified by CCK–8. (**C**–**F**) The proliferation of OSCC cells was suppressed by NF, as demonstrated by the colony formation assay (80 μM, 7 d) and EdU assay (80 μM, 24 h). (**G**) Increased expression of pro-apoptosis markers Bax and Caspase 3, and decreased expression of anti-apoptosis marker Bcl-2 in OSCC cells treated with NF (80 μM, 48 h) were detected by western blot. (**H**,**I**) The apoptosis-promoting effects of NF (80 μM, 48 h) on OSCC cells were verified by flow cytometry. ns, not significant. * *p* < 0.05, ** *p* < 0.01, *** *p* < 0.001 and *** *p* < 0.0001.

**Figure 2 ijms-24-14532-f002:**
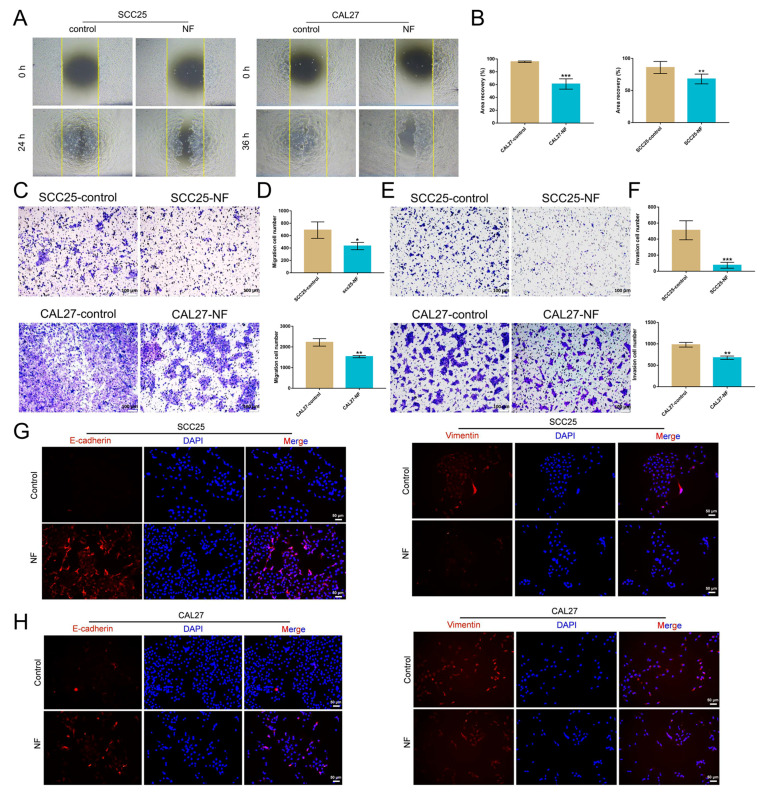
NF inhibited the migration and invasion of OSCC cells. (**A**,**B**) The migration of OSCC cells was inhibited by NF at a non-lethal concentration (30 μM, 24 h or 36 h), as confirmed by the wound healing assay. (**C**,**D**) Migration of OSCC cells was suppressed by NF at a non-lethal concentration (30 μM, 24 h), as demonstrated by a transwell assay. (**E**,**F**) The invasion-inhibiting effects of NF of a non-lethal concentration (30 μM, 24 h) on OSCC cells were tested by a transwell system. (**G**,**H**) Increased expression of E-cadherin and decreased expression of Vimentin in OSCC cells after NF treatment (80 μM, 48 h) were detected by immunofluorescence. * *p* < 0.05, ** *p* < 0.01 and *** *p* < 0.001.

**Figure 3 ijms-24-14532-f003:**
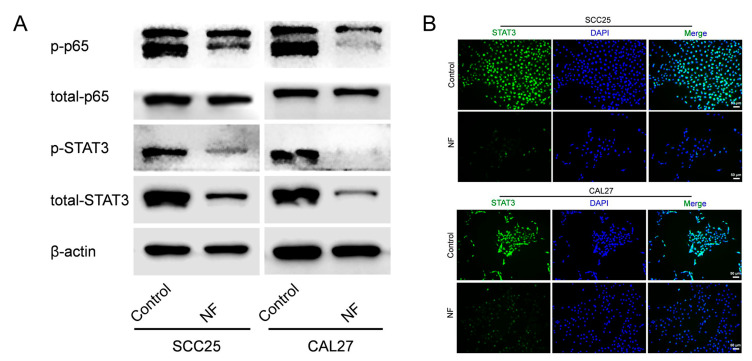
STAT3 signaling pathway was suppressed in NF-treated OSCC cells. (**A**) p65 and STAT3 signaling pathways were suppressed in OSCC cells treated with NF (80 μM, 48 h). (**B**) Suppressed STAT3 signaling pathway in NF-treated OSCC cells (80 μM, 48 h) was confirmed by immunofluorescence.

**Figure 4 ijms-24-14532-f004:**
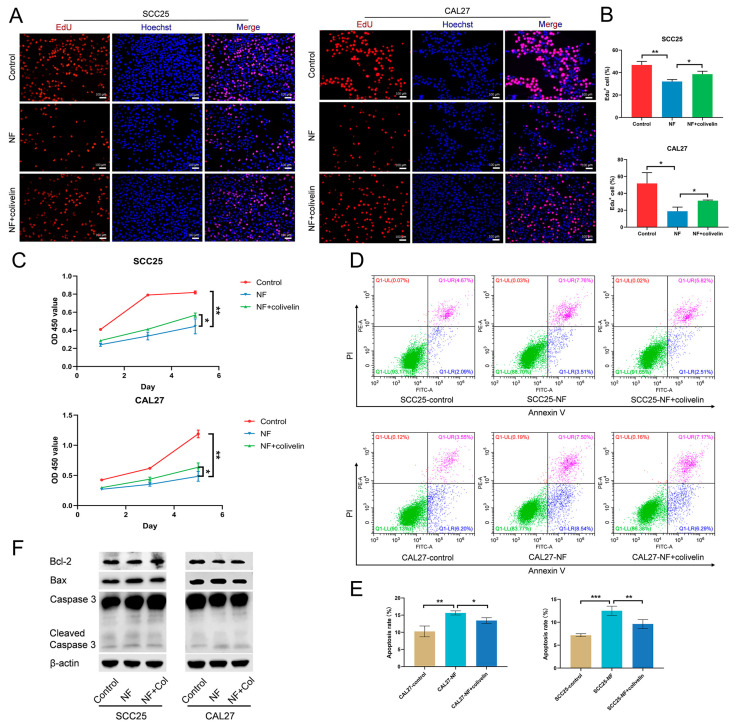
NF inhibited the proliferation and promoted the apoptosis of OSCC cells via suppressing the STAT3 signaling pathway. (**A**,**B**) NF-suppressed (80 μM) OSCC cell proliferation was promoted by colivelin (10 μM), as demonstrated by EdU assay (24 h). (**C**) NF-suppressed (80 μM) OSCC cell proliferation was promoted by colivelin (10 μM), as demonstrated by CCK-8 test (1, 3, 5 d). (**D**,**E**) The apoptosis-promoting effects of NF (80 μM) were weakened by colivelin (10 μM), as demonstrated by flow cytometry (48 h). (**F**) Increased expression of pro-apoptosis markers Bax and Caspase 3, and decreased expression of anti-apoptosis marker Bcl-2 in NF-treated (80 μM) OSCC cells was reversed, as detected by western blot (48 h). Col, colivelin. Scale bar, 100 μm. * *p* < 0.05, ** *p* < 0.01 and *** *p* < 0.001.

**Figure 5 ijms-24-14532-f005:**
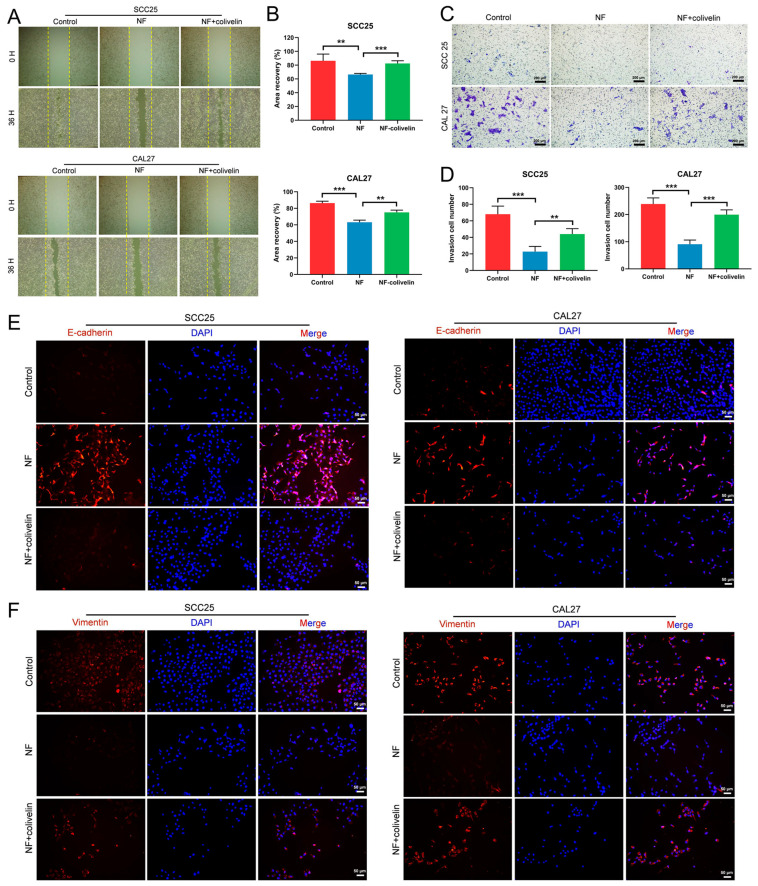
NF inhibited the migration and invasion of OSCC cells via suppressing the STAT3 signaling pathway. (**A**,**B**) NF-suppressed (30 μM) OSCC cell migration was promoted by colivelin (10 μM), as confirmed by the wound healing assay (24 h). (**C**,**D**) The invasion-inhibiting effect of NF (30 μM) on OSCC cells was weakened by colivelin (10 μM), as tested by a transwell system. (**E**,**F**) Increased expression of E-cadherin and decreased expression of Vimentin in NF-treated (80 μM) OSCC cells were reserved by colivelin (10 μM), as detected by immunofluorescence (48 h). ** *p* < 0.01, *** *p* < 0.001.

**Figure 6 ijms-24-14532-f006:**
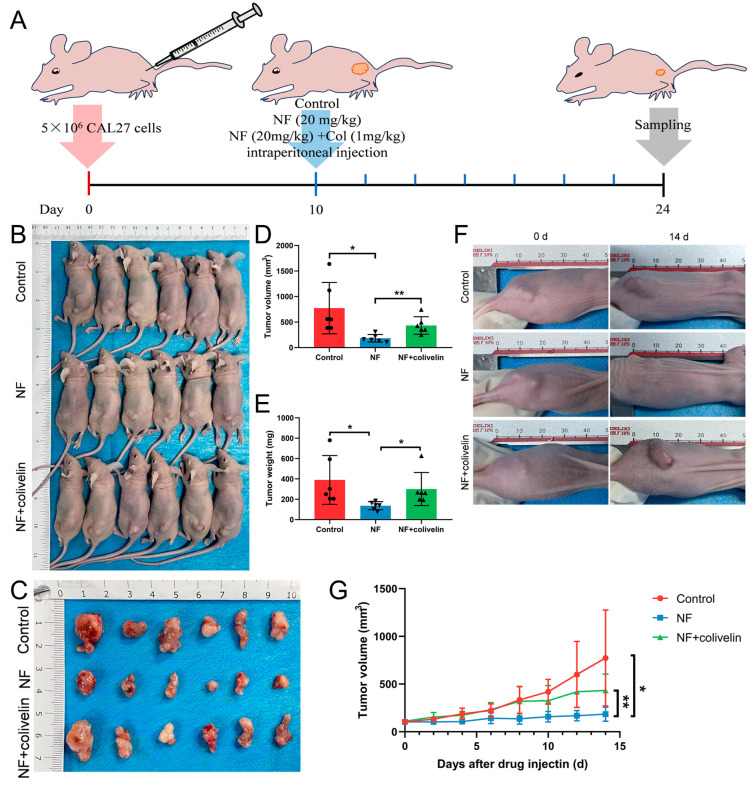
NF inhibited OSCC growth via suppressing the STAT3 signaling pathway. (**A**) The experimental procedure of CDX model. (**B**,**D**) Gross observation and tumor volume measurement of the mice in 3 groups after 2 weeks of drug injection. (**C**,**E**) Gross observation and tumor weight measurement of the isolated tumors in 3 groups after 2 weeks of drug injection. (**F**,**G**) Dynamic recording of tumor volume in mice in 3 groups during the process. * *p* < 0.05, ** *p* < 0.01.

**Figure 7 ijms-24-14532-f007:**
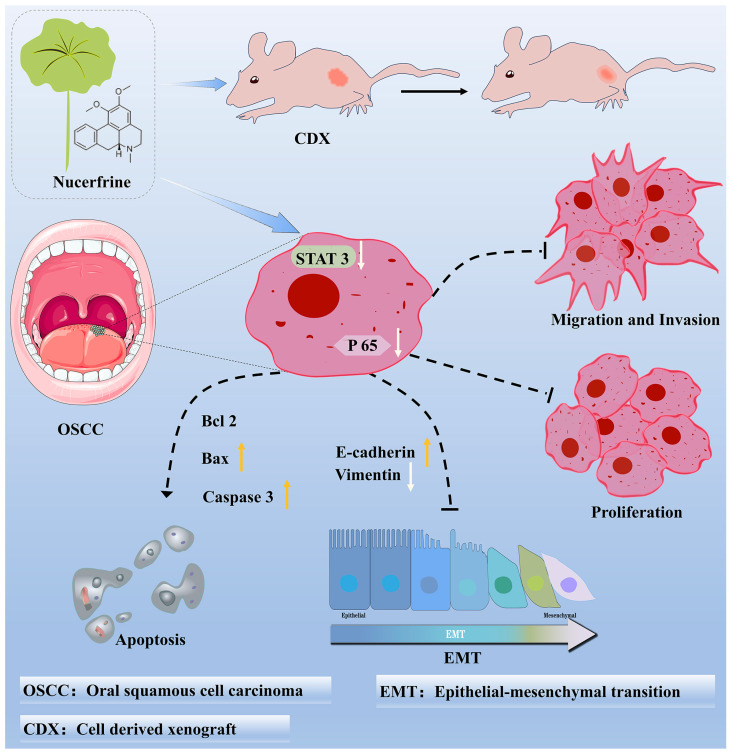
Illustration diagram of the study. In vitro, nuciferine inhibited the proliferation, migration and invasion, and promoted the apoptosis of OSCC cells. In vivo, nuciferine significantly inhibited tumor growth in the CDX model. Mechanismly, nuciferine functioned the anti-OSCC effects partially through suppressing the STAT3 signaling pathway.

## Data Availability

Data is contained within the article or Appendix A.

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
