# Peer review of "Nuciferine Inhibits Oral Squamous Cell Carcinoma Partially through Suppressing the STAT3 Signaling Pathway"

_ijms, 2023, doi:10.3390/ijms241914532_

Round 1
Reviewer 1 Report
In the submitted manuscript, the authors study the therapeutic activity of nuciferin in Oral squamous cell carcinoma (OSCC). The authors showed that nuciferine inhibited the growth, movement, and infiltration of OSCC cells, and also increased the programmed cell death of OSCC cells in vitro. Treatment with nuciferine suppressed the STAT3 signaling pathway, and the activation of the STAT3 signaling pathway by colivelin reversed the effect of nuciferine treatment of OSCC cells. Therapeutic effect of nuciferine was demonstrated in vivo, using the cell-based xenograft (CDX) model in nude mice. Colivelin diminished the tumor-inhibiting impact of nuciferine, suggesting that nuciferine might partially exert its therapeutic effect OSCC by suppressing the STAT3 signaling pathway. The authors conclude that nuciferine could offer a fresh alternative for the pharmaceutical management of OSCC.
Determining the novel therapeutic modalities against OSCC is an important topic. Results are novel and have a translational impact for treatment of OSCC. Thus, these results are of high importance to the broad readership of the International Journal of Molecular Sciences (IJMS). Most of the experiments were well executed with a very little flaw. However, there are several minor concerns that should be addressed in order for the manuscript to be suitable for publication in IJMS: 1) Treatment with nuciferine showed a good in vivo therapeutic effect, but in vitro, it required very high doses of nuciferine (80 microM) to achieve the therapeutic effect. The authors should address this in discussion; 2) Due to the high doses of nuciferine (80 microM) and colivelin (10 microM) used for in vitro experiments, it is possible that the presented results obtained in vitro, might not reflect the in vivo effect of nuciferine – this should be clearly stated in discussion.
Thus, due to the high novelty and translational significance, this manuscript has a high potential to significantly advance the field. Addressing the concerns outlined above would make the manuscript more complete and suitable for a broad readership of the IJMS.
Reviewer 2 Report
Since cancer chemotherapy drugs have been showing lot of side effects, authors in the present study have investigated antitumor effects alternative drugs that are safe derived from the plants. In the current study authors used Nuciferine, a bioactive compound from the lotus leaves, and found that Nuciferine exhibits antitumor effect by suppressing the STAT3 signaling in OSCC.
1. Although chemotherapy drugs are having lot of side effects, its efficacy is very good compared to natural drugs. It would have been better if the authors were used the one of the chemotherapy drugs as positive control or combine Nuciferine with chemotherapy drugs to improve the efficacy while decreasing the side effects by decreasing dose of chemotherapy drugs.
2. There is no Figure 1A (cytotoxicity assay).
3. In Fig-4f (western blot) authors showed that there was decrease in BCL-2 protein expression in nuciferine group compared to control but as per image there was not much difference in band size. It is better to show the band intensity graphs for western blots.
